# Opportunities for better value wound care: a multiservice, cross-sectional survey of complex wounds and their care in a UK community population

Trish A Gray,[1,2] Sarah Rhodes,[2,3] Ross A Atkinson,[1,2] Katy Rothwell,[2] Paul Wilson,[2,4] Jo C Dumville,[1,2] Nicky A Cullum[1,2,5]

[1]Division of Nursing, Midwifery and Social Work, School of Health Sciences, Faculty of Biology, Medicine and Health, University of Manchester, Manchester, UK
[2]NIHR CLAHRC Greater Manchester, Salford Royal NHS Foundation Trust, Salford, UK
[3]Centre for Biostatistics, Faculty of Biology, Medicine and Health, University of Manchester, Manchester, UK
[4]Alliance Manchester Business School, University of Manchester, Manchester, UK
[5]Research and Innovation Division, Manchester University NHS Foundation Trust, Manchester, UK

**Correspondence to**
Dr Trish A Gray;
trish.gray@manchester.ac.uk

## ABSTRACT

**Background** Complex wounds impose a substantial health economic burden worldwide. As wound care is managed across multiple settings by a range of healthcare professionals with varying levels of expertise, the actual care delivered can vary considerably and result in the underuse of evidence-based interventions, the overuse of interventions supported by limited evidence and low value healthcare.

**Objectives** To quantify the number, type and management of complex wounds being treated over a two-week period and to explore variations in care by comparing current practices in wound assessment, prevention and treatment.

**Design** A multiservice cross-sectional survey.

**Setting** This survey spanned eight community services within five Northern England NHS Trusts.

**Results** The point prevalence of complex wounds in this community-based population was 16.4 per 10 000 (95% CI 15.9 to 17.0). Based on data from 3179 patients, antimicrobial dressings were being used as the primary dressing for 36% of patients with complex wounds. Forty per cent of people with leg ulcers either had not received the recommended Doppler-aided Ankle Brachial Pressure Index assessment or it was unclear whether a recording had been taken. Thirty-one per cent of patients whose most severe wound was a venous leg ulcer were not receiving compression therapy, and there was limited use of two-layer compression hosiery. Of patients with a pressure ulcer, 39% were not using a pressure-relieving cushion or mattress.

**Conclusions** Marked variations were found in care, underuse of evidence-based practices and overuse of practices that are not supported by robust research evidence. Significant opportunities for delivering better value wound care therefore exist. Efforts should now focus on developing strategies to identify, assess and disinvest from products and practices supported by little or no evidence and enhance the uptake of those that are.

## INTRODUCTION

Complex wounds, (wounds with superficial, partial or full-thickness skin loss healing by secondary intention) such as lower limb diabetic or venous ulcers, pressure ulcers,

### Strengths and limitations of this study

► This cross-sectional survey provides robust community-focused population point prevalence estimates for different types of complex wounds.
► This is the first community-focused multiservice survey to capture the wide variation in treatment and care of complex wounds between different National Health Service trusts.
► The survey is based on wound care provided by community services and may have missed patients only treated by other service providers such as acute or primary care or those self-treating.
► As this was an anonymised survey, we were unable to conduct any case validation or validate wound aetiology.

open trauma and surgical wounds,[1 2] impose a substantial health economic burden worldwide. In the UK, the point prevalence of complex wounds is estimated at 14.7 per 10 000 population, suggesting that approximately 80 000 people in the UK have one or more complex wound at any one time.[3]While the annual cost of managing wounds has been crudely estimated at £3 billion in the UK,[4] US$2.85 billion in Australia[5] and US$25 billion in the USA,[6] the true cost is unknown. The increasing prevalence of complex wounds with age[7] and multimorbidity[8] means that it is difficult to separate the cost of wound care from the cost of caring for people with complex needs.

Wound care is managed across multiple settings by a range of healthcare professionals (HCPs) with varying levels of expertise.[9–13] HCPs are constantly under pressure to make the right choices for their patients, but when faced with an array of wound care products to choose from,[14] inaccessible or limited research evidence to guide decisions[9–14] and silo-based cost-control measures,[15] decisions

BMJ

may not be based on best practice[16 17] and fragmentation of practice and services may occur.[18] With such diversity, it is unlikely that all patients with wounds will have access to good value healthcare.[19] Unwarranted variation in healthcare at a time of rising demand is a concern for health systems globally. It has been estimated that around a third of medical practices are effective or likely to be effective, 50% are of unknown effectiveness and 15% are harmful or unlikely to be beneficial.[20] The cost-effectiveness[21] and value are even less well known.

Value in healthcare is centred on the interests and activities of all stakeholders and is measured by outcomes achieved relative to cost. This perspective shifts the focus on the delivery of care from volume alone to the value gained for patients from the healthcare investments being made, for example, staff time and treatment costs.[19] Every individual should have access to value-based care that considers their needs, preferences and priorities.[19] Gaining maximum healthcare value, given available resources, has been central to discussions and policy changes worldwide in recent years.[22] Decision-makers, especially those at policy and organisational levels, are increasingly aiming to maximise patient benefit while minimising the opportunity costs of current healthcare approaches. The term 'opportunity cost' refers to the potential for used resources to achieve more value elsewhere in the healthcare system.[23] High opportunity costs occur with the overuse of ineffective treatments (or those that make a very small clinical difference), leading to wasted patient benefit and reduced value in the healthcare system. Conversely, underuse of treatments known to be effective also leads to waste.[22]

In some cases, large improvements in the value of healthcare can be made relatively easily by the implementation of evidence-based guidelines[14 24 25] and smarter procurement of services and products.[26] We are seeing the development of initiatives focused on reducing variation and improving value in the UK[23 26–28] and across the world.[29–31] One campaign launched in 2012, 'Choosing Wisely', has been adopted by 12 countries.[32]

The first step to reducing unwarranted variation is to identify (1) overuse of interventions that do not clearly offer value and (2) underuse of interventions known to offer value. Once such situations have been identified, organisations can start to work towards increased use of effective practices across the relevant populations they are responsible for. To date, there have been no such initiatives focused in wound care; a significant yet often neglected area of care and healthcare spend, where there is great potential for better value care. To this end, we explored variation in common interventions for complex wounds across eight community services (spanning five National Health Service (NHS) Trusts) in the north of England. This project forms part of a wider wound care programme developed by the National Institute for Health Research Collaboration for Leadership in Applied Health Research and Care Greater Manchester (NIHR CLAHRC GM). Given the dearth of basic epidemiological data on

chronic or complex wounds in community settings,[33–35] we sought to quantify the number, type and management of complex wounds being treated over a two-week period. We then compared current practices in wound assessment, prevention and treatment with evidence-based recommendations, allowing exploration of variations in care between areas where distinct healthcare organisations provide care to geographically proximal residents. Key treatments of interest included the use of antimicrobial wound dressings, compression for people with venous leg ulcers (VLUs) and the use of pressure relief for pressure ulcer prevention, as these are specifically highlighted in recent guidelines,[14 24 36–38] as presented in table 1.

## MATERIALS AND METHODS
### Project design and participating organisations
A multiservice, cross-sectional survey recorded wound prevalence and care for people living with complex wounds across eight community services within five NHS Trusts in the north of England (with a population of 1.9 million). The methods used were based on a previous multiservice complex wound survey undertaken in Leeds, UK.[3]

As most wound care takes place in the community, the project focused on community-based wound care and did not include hospitals or primary care. HCPs working in a range of services (including tissue viability, adult and children's community nursing, podiatry, intermediate care, burns and plastics, specialist leg ulcer and specialist diabetic foot teams,) collected data for consecutive patients over a period of two weeks at each site between June 2015 and September 2016. Face-to-face training and accompanying instructions were provided to improve data accuracy and conformity of responses. Reminders were provided via telephone and email in the week leading to commencement. All services had a helpline number they could call if they had any questions either before or during the process. This survey was defined as service review by the Trusts' Research and Development Departments and did not require approval from a research ethics committee.

### Data collection
A survey was developed (online supplementary appendix 1), largely based on one used successfully in a similar study.[3] The questionnaire was designed to capture the number and nature of wounds and the care provided in terms of workforce, service configuration, assessments used (ie, Ankle Brachial Pressure Index (ABPI)), treatment choices and product usage. Survey forms were delivered to all clinical areas across the five community NHS Trusts. Data were collected for all patients receiving one or more episodes of wound care for their complex wound(s) from an NHS community service during a two-week period. Forms were completed following the consultation during office time, without the patient present and were

**Table 1**  Key evidence and recommendations related to the assessment, prevention and treatment of complex wounds

| Wound | Key treatments of interest | Guidelines and recommendations |
|---|---|---|
| Infected (All complex wounds) | Silver dressings | Insufficient evidence to support the use of silver-containing dressings to promote wound healing or prevent wound infections[14] |
| | Honey | Some high-quality evidence (based on two RCTs only) has shown honey to heal partial thickness burns and infected postoperative wounds more quickly than comparators; however, comparators may not be relevant to current practice. Insufficient evidence to support the use of honey in other wounds[14] |
| | Iodine | There is insufficient evidence addressing effectiveness and safety for use of iodine to treat or prevent wound infection.[14] |
| VLU | ABPI | Measurement of ABPI should be performed by appropriately trained practitioners to substantiate the presence or absence of PAD at initial assessment and to regularly review the use of compression therapy.[37] |
| | Compression therapy | Simple non-adherent dressings and high-compression multicomponent bandaging should be used for treating patients with VLU and ABPI ≥0.8. Graduated compression hosiery is recommended to prevent recurrence of VLU.[37] Two-layer compression stockings are as clinically effective as high-compression bandages but more cost-effective[35] |
| | Pentoxifylline | High-quality evidence, based on systematic review and meta-analysis has found improved VLU healing with the use of pentoxifylline (believed to increase microcirculatory blood flow although exact mechanism of action is unknown) and should be considered in patients with VLU.[37 49] |
| Diabetic foot ulcer | Dressing choice | Insufficient evidence to support the use of any specific dressing. Clinical assessment and patient preference should be taken into consideration, while the lowest acquisition cost appropriate to the clinical circumstances should be used.[14 36] |
| | Pressure relief | Offer non-removable casting to offload plantar neuropathic, non-ischaemic, uninfected forefoot and midfoot diabetic ulcers taking into consideration clinical assessment and patient preference. Use pressure redistributing devices and strategies to minimise the risk of pressure ulcers developing.[36] |
| PU | Dressing choice | Insufficient evidence to support the use of any specific dressing, choice should be determined by the patient's pain, tolerance, location of the ulcer and amount of exudate. A dressing that promotes a warm, moist wound-healing environment should be considered for grades 2, 3 and 4 PUs.[14 24] |
| | Pressure relief | Use high-specification foam mattresses or consider the use of dynamic support surface if not sufficient. Consider high-specification foam or equivalent pressure redistributing cushion for chair or wheelchair use.[24] |

ABPI, Ankle Brachial Pressure Index; PAD, peripheral arterial disease; PU, pressure ulcer; RCTs, randomised controlled trials; VLU, venous leg ulcer.

anonymised at source. To reduce duplication, forms were completed by the person providing the most hands-on care. This was supported by further local processes, such as placing stickers in patients' clinical notes when a form had been completed. Potential duplicates were not removed from the dataset; as data were anonymised we could not be sure whether they were true duplications.

The survey consisted of structured questions relating to patient demographics, wound identification by type, number and severity. The form also asked for more detail about each patient's most severe wound. Thus, if a patent had multiple complex wounds, more detailed information (eg, on wound treatment) was only collected for one wound. For patients with multiple wounds of equal severity, HCPs were asked to focus on the largest

wound. Information requested relating to investigations undertaken (eg, Doppler-aided measurement of ABPI) or treatments provided (eg, providing pressure relieving equipment) are all specific aspects of care carried out in community settings by the HCPs (and their colleagues) responsible for completing the questionnaire, and it would be expected that such clinical procedures would be recorded in the patient's community health record. The survey had been used previously[3] and was piloted locally prior to commencement by a range of HCPs. Minor changes were made in the light of feedback received.

## Data analysis

All data were analysed using simple summary statistics; numbers with percentages for categorical data and mean/

median values with range for numerical data. Community point prevalence(CPP) rates per 10 000 population were produced for each wound type, along with 95% CI. The denominator used for these calculations was 1 935 683 based on total population figures for the five Trusts surveyed, taken from Health and Social Care Information Centre 2015 data.

## RESULTS
### Description of community complex wound population
Overall, we recorded data for 3179 patients with a total of 5632 complex wounds (median number of wounds per patient 1, range 1–24), corresponding to an overall CPP of 16.4 per 10 000 (95% CI 15.9 to 17.0). People with complex wounds tended to be elderly (median age: 74) with at least one comorbidity (median 1.0, range 0–9). Cardiovascular disease was the most frequently reported comorbidity (in 1808 patients; 57%) followed by diabetes (817; 26%) and arthritis (641; 20%). Just under a half of the patients with complex wounds were immobile or walked with difficulty (table 2) and 494 (31%) of 1613 patients reported as being fully mobile were receiving home visits from a HCP. VLUs were the most prevalent complex wound type (n=612; CPP 3.2 per 10 000; 95% CI 2.9 to 3.4), followed by diabetic foot ulcers (n=488; CPP 2.5 per 10 000; 95% CI 2.3 to 2.7), traumatic wounds (n=428; CPP 2.2 per 10 000; 95% CI 2.0 to 2.4) and pressure ulcers (n=348; CPP 1.8 per 10 000; 95% CI 1.6 to 2.0).

### Use of wound dressings for all complex wounds
Wound dressings are applied to all types of complex wound. Of the 3038 patients for whom data on the 'most severe wound' were provided, 1096 (36%) patients were receiving an antimicrobial primary dressing: 383 (13%) a silver dressing and 713 (23%) patients were receiving other antimicrobial dressings such as iodine or honey. There was marked variation in the use of antimicrobial dressings across community services as shown in figure 1, ranging from 18% of patients in one area to 69% in another. As noted in table 1, there is currently insufficient evidence to support the use of antimicrobial dressings to promote wound healing.[14] It is highly uncertain that antimicrobial dressings are clinically or cost-effective and there is no high-quality evidence that they improve wound healing, reduce infection rates or reduce the prescribing of systemic antibiotics.[14 39] For VLUs, the use of antimicrobial dressings is specifically not recommended.[37]

### Assessment and treatment of leg ulcers
A leg ulcer was reported as being the most severe wound in 25% (n=770) of patients with a most severe wound selected. For these patients a Doppler-aided ABPI measurement is a crucial part of the assessment process to rule out significant peripheral arterial disease and determine treatment choice and access (table 1).[37] In total, of those patients with one or

| Table 2 Demographic characteristics of patients with at least one complex wound | |
| --- | --- |
| **Characteristic** | |
| Gender (n=2967) | |
| Male: n (%) | 1439 (49) |
| Female: n (%) | 1528 (51) |
| Ethnicity (n=3152) | |
| White British: n (%) | 2819 (89) |
| Other: n (%) | 336 (11) |
| Age (n=3120) | |
| Median (range) | 74 (1–107) |
| Accommodation (n=3157) | |
| Own/rented home: n (%) | 2728 (86) |
| Nursing/residential home: n (%) | 348 (11) |
| Other: n (%) | 84 (3) |
| Number of comorbidities (n=3179) | |
| Median (range) | 1 (0–9) |
| Continence (n=3029) | |
| No incontinence: n (%) | 2487 (82) |
| Urinary or faecal incontinence or both: n (%) | 542 (18) |
| Mobility (n=3141) | |
| Fully mobile: n (%) | 1613 (51) |
| Walks with difficulty: n (%) | 1091 (35) |
| Immobile: n (%) | 437 (14) |
| Community point prevalence per 10 000 population for most common wound types: n (CPP; 95% CI) | |
| Venous leg ulcer | 612 (3.2; 2.9 to 3.4) |
| Diabetic foot ulcer | 488 (2.5; 2.3 to 2.7) |
| Traumatic wound | 428 (2.2; 2.0 to 2.4) |
| Pressure ulcer | 348 (1.8; 1.6 to 2.0) |

CPP, community point prevalence.

more leg ulcer, 19% (n=150) did not have an ABPI recorded in community-held notes. The frequency of ABPI recording varied across services from 10% to 28% and for a further 21% (n=167); this information was either unknown or not reported (ranging from 13% to 22%).

A VLU was reported as being the most severe wound in 570 patients of whom 175 (31%) were recorded as receiving no compression therapy, ranging from 2% to 30% (compression is an effective first-line treatment for venous ulcers[40 41]). Half (n=287; 50%) of those with VLUs were managed with compression bandages, 79 (14%) with compression hosiery and 29 (5%) with a combination of the two (figure 2). There was limited use of two-layer compression hosiery across all areas surveyed despite its known cost-effectiveness relative to compression bandaging.[35] None of

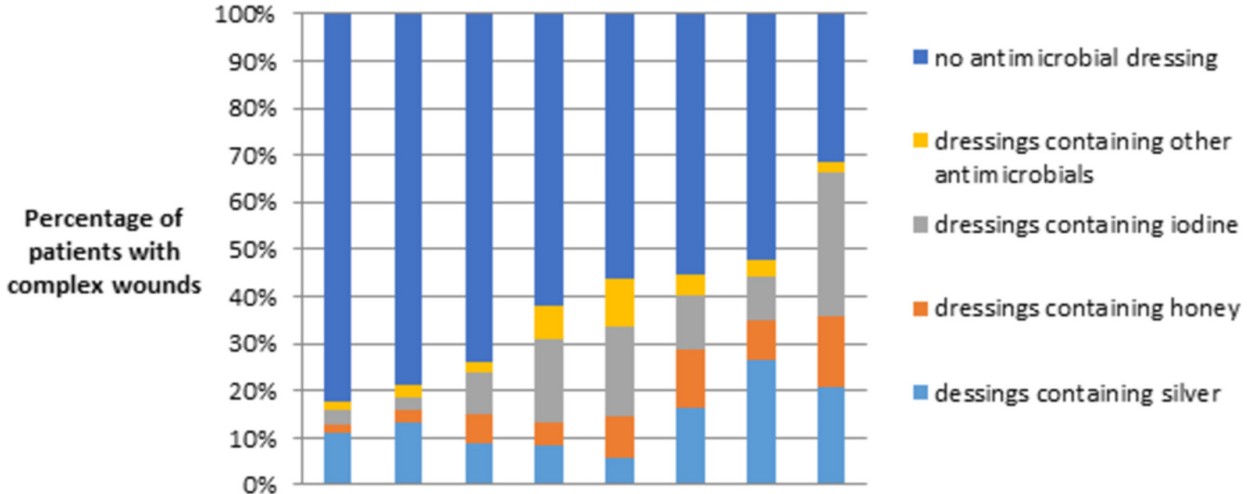

**Figure 1** Proportion of complex wounds for which primary dressing contained antimicrobials: the other antimicrobial dressing group maps to the same section of the British National Formulary and includes dressings such as polyhexanide polyhexamethylene biguanide (bars represent included community services). Number of patients per community service ranged from 172 to 655.

the patients with a diagnosis of VLU were prescribed pentoxifylline; a treatment shown to be clinically and cost-effective and recommended to promote the healing of VLUs.[37]

### Use of support surfaces in those at risk of pressure ulceration

Over one-third (35%) of patients with complex wounds were reported as being at risk of pressure ulceration and 348 (11%) had a pressure ulcer at the time of survey. Of the 281 patients whose most severe wound was a pressure ulcer (who are thus known to be at high risk of further ulceration and should be receiving pressure relief[24]), 109 (39%) patients were reported as not having a pressure-relieving cushion

or mattress (ranging from 27% in one area to 64% in another; figure 3). For the 711 patients with a foot ulcer selected as the most severe wound, 40% (n=286) were not receiving any pressure relief for the affected foot (ranging from 31% to 60%).

### DISCUSSION

This study characterised the number and nature of complex wounds being cared for by NHS community services and the assessments and treatments being used in their management. Our estimate of the point prevalence of complex wounds (16.4 cases per 10 000

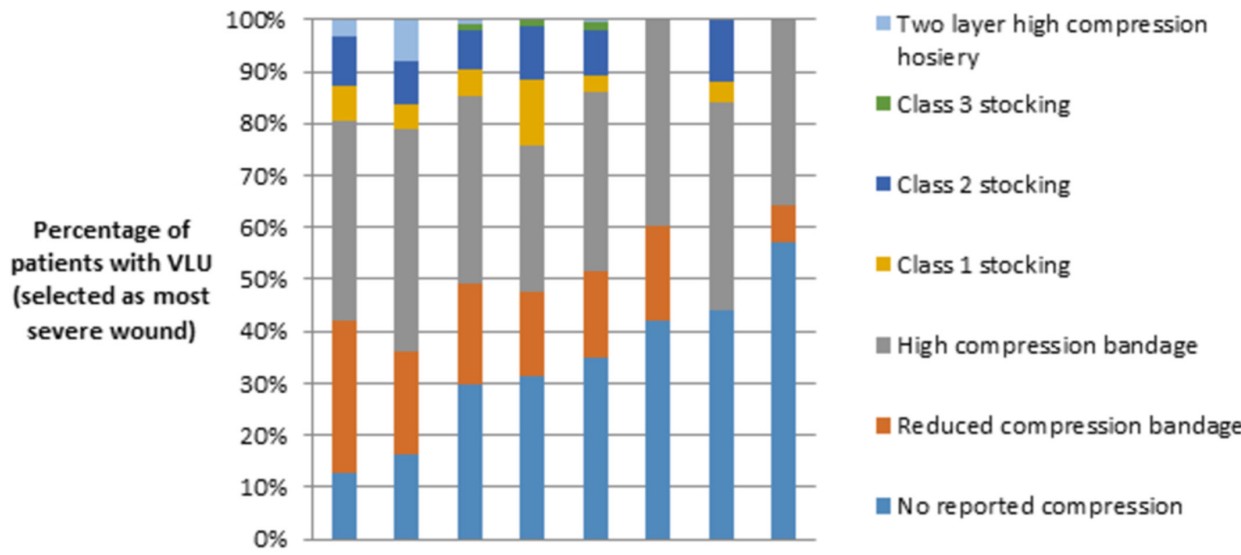

**Figure 2** Highest level of compression used for patients with VLUs selected as most severe wound (bars represent included community services). Number of patients per community service ranged from 14 to 151. VLU, venous leg ulcer.

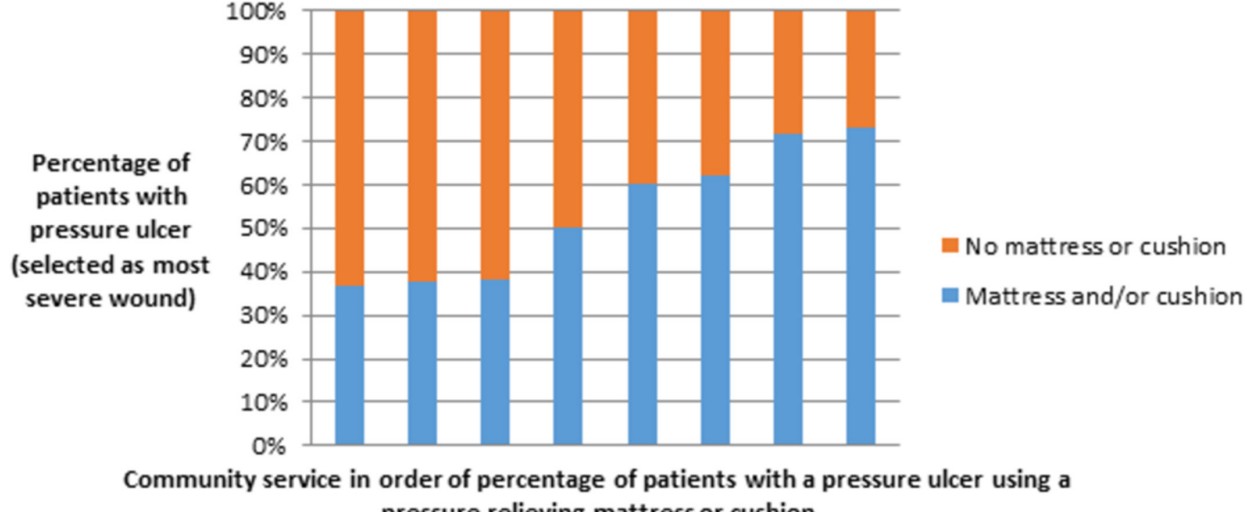

**Figure 3** Proportion of patients using pressure-relieving mattresses or cushions (bars represent included community services). Number of patients per community service ranged from 10 to 63.

population) is consistent with another estimate from the north of England (14.7 per 10 000 population[3]). As these are point prevalence estimates from general populations using census figures as denominators, the estimates are lower than those estimates which focus on specific high-risk populations (such as hospital patients).[1] We found indicators that revealed unwarranted variation in clinical practice across participating services; the underuse of evidence-based interventions (eg, compression therapy for venous ulcers) and the overuse of interventions supported by limited evidence (eg, antimicrobial dressings). Overuse of such treatments could incur opportunity cost, drawing resource from the system that cannot be used to confer benefit elsewhere.

### Overuse of interventions supported by limited evidence
Across community services, the survey revealed substantial variation in the use of antimicrobial dressings for wound management, ranging from 18% to 69% of all primary dressings used. The unit cost of antimicrobial dressings is often higher than that of their non-antimicrobial counterparts; their use is associated with a high cost for little or no known patient benefit. While one could postulate that the use of antimicrobial dressings might reduce the prescribing of systemic antibiotics (which would be highly desirable in the context of antimicrobial resistance[42]), there is no evidence that this is the case. To examine the use of antimicrobial dressings in more depth, we considered silver dressings for which current evidence on the relative effectiveness for preventing or treating wound infection and promoting healing is uncertain.[43 44] Silver dressings cost several times more than their non-silver counterparts. In a conservative scenario where silver dressings are used for a patient over a two-week period with two dressing changes per week, the dressing cost (using an average cost for a small dressing) would be approximately £32. The cost for a standard foam dressing (of the same size) in the same scenario is approximately

five times less (£6.80). Use of silver dressings in this situation (assuming all other aspects of care and outcome are equal) corresponds to an additional £2520 per 100 patients treated with silver dressings. Tackling overuse is not an easy task and not solely the responsibility of individual HCPs. But despite increased awareness among clinicians, policy-makers and the general public during the last 5–10 years, there have not been widespread changes in policy to address this issue.[45] In wound care specifically, removing silver dressings from a prescribing formulary would offer a partial solution only. There is a clear need for a rational strategy for identifying, assessing and disinvesting from products supported by little or no evidence; ideally, this should be delivered at a health system level.[46]

### Underuse of evidence-based interventions
Across our population, we saw underuse of ABPI in those with leg ulcers (with 19% not having this measure recorded and for 21% it was unclear whether a recording had been taken). Such underassessment (whether a delay or omission) is likely linked to the underuse of compression therapy also observed. These related issues require urgent attention since compression is a highly efficient and cost-effective treatment for VLU.[37] The first response to this finding is to seek an understanding of the reasoning behind the observed patterns of compression use. For example, underuse may be due to patients waiting for ABPI assessment or ABPI assessment identifying peripheral arterial disease that may preclude compression use. In other cases, lack of compression may reflect issues with patient adherence to therapy or a shortage of staff with compression bandaging skills. The extent to which current findings can be explained by appropriate contraindications is unclear and requires further investigation. Crucially, the reason for such wide variation between relatively local areas also needs further explanation.

Using data from randomised controlled trials (RCTs),[47] [48] we conservatively calculate that non-use of compression in those with VLUs will lead to 23 fewer people per 100 patients with VLU healing over 12 months compared with use of compression. Using an estimated mean cost per VLU episode of £1800,[35] we can estimate an annual increased cost of £41 400 per 100 patients with VLU not using compression. However, we could tentatively also consider non-use of compression as a proxy for suboptimal care. If we apply the recent Rightcare[28] estimated cost savings associated with optimised VLU care (including high compression use), savings could be as high as £118 979 per 100 patients with VLU. While here we have focused on cost impacts, it is important to consider the improved health-related quality of life associated with a healed wound as well as the reduced financial burden on patients, especially those who are unable to work or require time away from work to attend clinic appointments.

There is also an apparent underuse of pentoxifylline, a xanthine derivative used to treat muscle pain in people with peripheral arterial disease, but also believed to increase microcirculatory blood flow in patients with VLU. Uptake is low despite evidence from 12 RCTs included in a systematic review[49] that it increases ulcer healing both with and without compression. The reasons for this underuse are being explored further but are likely to include lack of awareness of the drug and its effects amongst both general practitioners (GPs) and community nurses, relative lack of nurse prescribers able to prescribe pentoxifylline and reluctance to prescribe for an off-licence indication (particularly in people who may be taking several other medicines for comorbidities). It is also worth noting that the drug is low cost but not actively promoted because the patent is long expired. It is unclear whether this treatment is used more widely outside the UK. Using published relative effectiveness estimates and related costs,[50] we have estimated that the use of pentoxifylline alongside compression could result in cost savings of over £40 000 per 100 VLU patients treated . Combined increases in the use of ABPI, cost-effective compression therapy and greater pentoxifylline use in the treatment of those with venous leg ulceration is likely to see a cumulative increase in improved patient outcomes, experiences and reduced healthcare costs.

Similar exploration is required in relation to the potential underuse of pressure relieving equipment suggested by our study to understand the true scope for improvement. In all cases, further work is required to identify the factors that underpin clinical decision-making and behaviour in these areas. An understanding of such factors would support the identification and selection of appropriate behaviour change techniques and implementation strategies, targeted at modifying these behaviours and generating improvement and thus value in healthcare. Having a pressure ulcer is prognostic for further pressure ulceration, thus those with an ulcer are considered at risk and should receive pressure-relieving interventions such as support surfaces. Even accounting for the initial cost of purchasing a device given the large cost associated with healing a pressure ulcer (estimated cost for healing a grade 2 pressure ulcer=£5241),[51] we conservatively calculate that the use of support surfaces to prevent pressure ulceration could save at least £38 000 per 100 at-risk patients. Again those with a pressure ulcer are also known to have worse health-related quality of life than others with similar comorbidities.[52] These are not isolated examples of underuse: a US study found that patients with acute or chronic conditions (that represented the leading causes of illness, death and use of healthcare) received only 55% of recommended care,[53] and similarly the CareTrack study found that Australians received appropriate care in only 57% of 35 573 eligible healthcare encounters.[54] However, this is the first work that has revealed so clearly, variation in the use of interventions for VLUs where there are corresponding guideline-based recommendations. Failure to deliver best practice is often a result of poor execution or lack of widespread adoption of best care processes.[55] International interest in research translation and quality improvement reflects the growing recognition of the slow and inconsistent uptake of effective healthcare practices worldwide.[56]

## Disinvestment and implementation of improvement initiatives to promote better value care

Local variation in product choice between organisations is shown clearly in the work presented. This highlights the huge scope for better value care in both the assessment and treatment of complex wounds; value could be released by disinvestment in some areas with savings being focused on areas of underuse identified here. Further work should be undertaken to understand the factors that underpin decision-making around treatment use, with a particular focus on exploring the motivations to use expensive treatments with limited evidence at a time when the NHS faces significant resource constraints. Working with our service partners, we plan to address the observed local practice variations through a programme of improvement work undertaken as part of NIHR CLAHRC GM. However, we recognise that this process needs to be replicated across the health system as a whole if overuse and underuse are to be addressed fully. As we have stated above, there is a clear need for a national strategy for identifying, assessing and disinvesting from products and practices supported by little or no evidence.[46] A model for taking a systematic approach to disinvestment already exists in the form of the Sustainability in Health care by Allocating Resources Effectively programme in Australia. This model enables clinicians, managers and policy-makers to manage the process from identifying the need for disinvestment to implementing the change and evaluating outcomes.[57] Similar efforts are required in the UK if practice variations are to be addressed.

## Strengths and limitations

This cross-sectional survey provides robust community-focused population point prevalence estimates for different types of complex wounds and is the first multiservice survey to capture the wide variation in treatment and care of complex wounds between different NHS trusts. There are a few limitations to our study. First, the study looked specifically at patients receiving care from NHS community services and did not include people whose wound care was delivered by hospitals, primary care or other care providers and also those who may have been self-caring at this time. While this prevents us from comparing the characteristics of people with and without complex wounds at the time of the survey, collection of these data would have been too resource intensive for participating services, and we know from previous work that the vast majority of people with wounds are cared for in community settings.[1] Second, we only scrutinised the patient information readily available to community staff and did not examine wider (eg, GP) patient records. However, given that we were looking for information regarded as crucial to the management of people with complex wounds, any omission of information from community records risks suboptimal clinical decision-making. Third, it is also conceivable that services failed to complete a survey form for each patient they saw with a complex wound during the survey period. However, both the level of engagement from services and the similarity between the estimated CPP obtained here and that found in a previous but smaller survey in Leeds[3] suggest the impact of this is likely to be negligible. Finally, as this was an anonymised survey we were unable to neither conduct any case validation nor validate wound aetiology. Consequently, data presented reflects the treating HCP's assessment.

## CONCLUSIONS

This survey adds important robust epidemiological data to the complex wound literature where existing prevalence data have been found to be limited when systematically reviewed.[1] We also highlight the overuse and underuse of services and treatments and reflect the issues emphasised in a number of reports relating to better value healthcare.[19 26 29] These findings suggest significant opportunities for delivering better value wound care exist. Efforts should now focus on developing strategies to identify, assess and disinvest from products and practices supported by little or no evidence and enhance the uptake of those that are.

**Correction notice** This article has been corrected since it first published. The word 'and' has been removed from the name of the author in the 'Correspondence to' section.

**Acknowledgements** The authors are grateful to healthcare professionals from NHS partner organisations for their time during data collection and their enthusiastic support throughout this project. The authors would also like to thank Wanda Russell (NIHR CLAHRC GM Research Fellow at the time of this project) for her role in conducting stakeholder consultation with practitioners and data collection and Louise Hussey, Research Fellow, for her contribution to data analysis.

**Contributors** NAC and JCD conceived the idea and design for the overall project; TAG, KR and PW contributed to further development of the study design; TAG and KR conducted stakeholder consultation with practitioners; SR was responsible for data analysis; RAA and JCD contributed to data analysis. TAG created the original draft of the manuscript. All authors contributed to the interpretation of study findings, critical revision of the manuscript for important intellectual content and approval of the final manuscript.

**Funding** This project was funded by the National Institute for Health Research Collaboration for Leadership in Applied Health Research and Care (NIHR CLAHRC) Greater Manchester. The NIHR CLAHRC Greater Manchester is a partnership between providers and commissioners from the National Health Service (NHS), industry and the third sector, as well as clinical and research staff from the University of Manchester.

**Disclaimer** The views expressed in this article are those of the authors and not necessarily those of the NHS, NIHR or the Department of Health and Social Care.

**Competing interests** None declared.

**Patient consent** Not required.

**Provenance and peer review** Not commissioned; externally peer reviewed.

**Data sharing statement** Requests for access to data should be addressed to the corresponding author.

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
