## [Reviewer comments · BMJ Open]

ARTICLE DETAILS

TITLE (PROVISIONAL)	OPPORTUNITIES FOR BETTER VALUE WOUND CARE: a multi-service, cross sectional survey of complex wounds and their care in a UK community population
AUTHORS	Gray, Trish; Rhodes, Sarah; Atkinson, Ross; Rothwell, Katy; Wilson, Paul; Dumville, Jo C.; Cullum, Nicky

VERSION 1 – REVIEW

REVIEWER	Perry Mayer The Mayer Institute Canada
REVIEW RETURNED	04-Oct-2017

GENERAL COMMENTS	Excellent review of current local wound care practice and cost-effectiveness.
---

REVIEWER	Rut F Öien Blekinge Center of Competence, Karlskrona, Sweden Faculty of Medicine, Lund University, Malmö, Sweden
REVIEW RETURNED	13-Nov-2017

GENERAL COMMENTS	The article displays an area of the highest interest to patients, staff and politicians. The survey is well organised and carried out but my deepest concern is that the key evidence and recommendations presented in Box 1 are not sufficient for suggesting significant opportunities for delivering better value wound care. As we know only 13% of the costs for wound management stands for dressing materials (and 87% for the time staff spend on wound management). It thus seems awkward to concentrate on some dressing materials to initiate better value wound care. For this purpose, it would have been better to focus on giving the patients the proper diagnosis and appropriate treatment strategies. Your ambition is good, to develop strategies to identify, assess and disinvest from products and practices supported by little or no evidence, and enhance the uptake of those that are. However, this article only doesn't reach your goal. Is the research question or study objective clearly defined? The definition of complex wounds (leg, foot and pressure ulcers which heal by secondary intention) should be more elaborated. Is there a time aspect (ulcers which have not healed within...?) Which aetiologies are included in the definition and so forth? You mention traumatic ulcers in Table 1 but not in Introduction. Please give a more detailed definition of "complex wounds".
---

I find the key word Patient safety not appropriate as it is not discussed any further in the article. I suggest it should be omitted. Why introducing the term of opportunity costs when it is not discussed any further?

Box 1 is quite intriguing to me.

In clinical practice antimicrobial dressings are not used to promote ulcer healing but is used when there are signs of local ulcer infection and then for a short time (10-14 days).

The use of antimicrobial dressings could also be one way to prevent irrational oral antibiotic treatment, which is a much larger threat to both the patients and society as a whole. It is clear that the use of antimicrobial dressings must be discussed in this context. What are your ideas concerning this issue?

Why have you omitted Polyhexanide (polyhexamethylene biguanide, PHMB)?

Local antibiotic therapy is mentioned. These agents are usually not recommended in wound management. Why have you mentioned them and which agents do you mean by local antibiotic therapy.

Why have you omitted flat-knitted stockings for good compression?

Pentoxifylline is not recommended for wound management in northern Europe and seems to be one agent that there is no consensus about. The reference is rather old (from 2010). Do you have further, more recent references, to back up your statement. In box 1, you clearly mention that this agent is believed to increase microcirculatory blood flow and should be considered in patients with VLU. Please elaborate your statement or simply omit this agent.

Is the study design appropriate to answer the research question?

The study period is very short (2 weeks) but I think it is sufficient to answer your questions.

A major concern is your statements about antimicrobial dressings and not relating these dressings to the overuse of oral antibiotic treatment, which is far more harmful to both the patients and society as a whole. I think you should relate these topics to one another and discuss this matter in detail.

Materials and Methods/Data collection

"To reduce duplication, discussions took place with services providing shared care to minimise the chance of data being collected twice for the same person". How exactly did you do that?

Do the results address the research question or objective?

The fact that you did find more men than women is intriguing as women usually are overrepresented in this kind of patient population. It would be a good idea to elaborate this somewhat surprising finding in Discussion.

Are the results presented clearly?

There seems to be room for better presentation of the results, like presenting your finding of the gender perspective in your material.

Are the discussion and conclusions justified by the results?

I think you have to rewrite the paragraph on silver as your assumptions for calculations are completely wrong. In clinical practice it is recommended to use silver dressings for a short period of time (10-14 days) for local ulcer infections. Silver dressings should only be prescribed by a physician. Dressing changes are usually made 2 a week. This makes your calculations way too high and not at all reflecting clinical practice. My suggestion is that you rewrite the whole paragraph on silver as it is quite misleading.

	Concerning compression therapy I do agree that there is much to be done within this field. I do miss a discussion about dressing changes, that is the use of the nurse's time, which stands for 87% of costs for wound management. With a proper compression therapy dressing changes could be done once a week, which very much reflects cost savings aspects. What is your comment on this? The largest concern is about pentoxifylline, which is scarcely used in northern Europe. You state that..."There is also an apparent underuse of pentoxifylline, a xanthine derivative used to treat muscle pain in people with peripheral arterial disease, but also believed to increase microcirculatory blood flow in patients with VLU". I do not think that this substance should be part of this article at all (only one reference from 2010). I advise you to omit this agent from the article. I do agree that in all cases, further work is required to identify the factors that underpin clinical decision making and behaviour in these areas. Opposed to you I do think that giving the patient a proper diagnosis and hence an appropriate treatment strategy would be a better way to achieve improvement and thus value in healthcare. This includes a shift of paradigm in wound management not based on dressing materials.
--	---

VERSION 1 – AUTHOR RESPONSE

Reviewer 1: Excellent review of current local wound care practice and cost-effectiveness.

Response: Thank you for taking the time to review the above manuscript and for the positive comments regarding our work.

Reviewer 2: The survey is well organised and carried out but my deepest concern is that the key evidence and recommendations presented in Box 1 are not sufficient for suggesting significant opportunities for delivering better value wound care. As we know only 13% of the costs for wound management stands for dressing materials (and 87% for the time staff spend on wound management). It thus seems awkward to concentrate on some dressing materials to initiate better value wound care. For this purpose, it would have been better to focus on giving the patients the proper diagnosis and appropriate treatment strategies. Your ambition is good, to develop strategies to identify, assess and disinvest from products and practices supported by little or no evidence, and enhance the uptake of those that are. However, this article only doesn't reach your goal.

Response: Thank you for this important point. As a general comment we agree that better service planning and provision will support optimal use of staff resourcing in the delivery of wound-related care. However, we don't wish to discount the opportunities for better value care from optimal treatment use. Indeed for some treatment such as compression for VLUs, better use may reduce healing times and that would in turn impact on staff time. Assessment of service planning and provision is a complex area that was beyond the remit of this study. We do feel that assessing the provision of ABPI assessment and compression and other analyses do address the area of 'appropriate treatment strategies'. Overall we feel that the reviewer's dismissal of the article as 'not reaching its goal' is inaccurate. In this manuscript we have focussed on those aspects where unwarranted variation in practice is revealed and where opportunities for delivering better value wound care exist. We accept that no one study could deal with the many challenges of maximising better value wound care overall.

Reviewer 2: The definition of complex wounds (leg, foot and pressure ulcers which heal by secondary intention) should be more elaborated. Is there a time aspect (ulcers which have not healed within...?) Which aetiologies are included in the definition and so forth?

Response: Thank you, we have revised to include more detail and added a reference (Page 4; line 78): Complex wounds (wounds with superficial, partial or full thickness skin loss healing by secondary intention), such as lower limb diabetic or venous ulcers, pressure ulcers, open trauma and surgical wounds.

Reviewer 2: You mention traumatic ulcers in Table 1 but not in Introduction. Please give a more detailed definition of “complex wounds”.

Response: We have included traumatic wounds to the definition as above.

Reviewer 2: I find the key word Patient safety not appropriate as it is not discussed any further in the article. I suggest it should be omitted.

Response: We agree and have removed ‘patient safety’ from key word list

Reviewer 2: Why introducing the term of opportunity costs when it is not discussed any further?

Response: We have added text to the discussion (Page 12; line 247): Overuse of such treatments could incur opportunity cost; drawing resource from the system that cannot be used to confer benefit elsewhere.

Reviewer 2: Box 1 is quite intriguing to me. In clinical practice antimicrobial dressings are not used to promote ulcer healing but is used when there are signs of local ulcer infection and then for a short time (10-14 days).

The use of antimicrobial dressings could also be one way to prevent irrational oral antibiotic treatment, which is a much larger threat to both the patients and society as a whole. It is clear that the use of antimicrobial dressings must be discussed in this context. What are your ideas concerning this issue?

Response: We thank the review for this thought-provoking comment. We have noted our thoughts below. We haven't made any changes to Box 1 as these are objective statements taken from Guidelines and reflect the current state of (un)certainty. Firstly, we believe that these dressings are not just used when there are signs of infection but also when “critical colonisation” is suspected e.g. something that cannot be easily diagnosed but often surmised by slow or non-healing.(1) Secondly, the underlying premise of the use of antimicrobial dressings is that by dealing with colonisation/infection, healing will proceed so fundamentally they are used to promote healing. The lack of clarity and consistency of antimicrobial dressing use is borne out by the variation we show in this paper.

(1) Wounds UK. Appropriate use of silver dressings in wounds: An expert working group consensus. 2012 http://www.woundsinternational.com/media/issues/567/files/content_10381.pdf

Reviewer 2: Why have you omitted Polyhexanide (polyhexamethylene biguanide, PHMB)?

Response: This dressing was considered in the other antimicrobial dressing category. All the dressings here mapped to the corresponding section of the UK British National Formulary ‘other antimicrobial dressing’ section – we have noted this in the figure legend (Page 17; line 377): Figure 1. Proportion of complex wounds for which primary dressing contained antimicrobials: the other antimicrobial dressing group maps to the same section of the British National Formulary and includes dressings such as Polyhexanide polyhexamethylene biguanide (bars represent included community services), Number of patients per community service ranged from 172 to 655

Reviewer 2: Local antibiotic therapy is mentioned. These agents are usually not recommended in wound management. Why have you mentioned them and which agents do you mean by local antibiotic therapy.

Response: This was taken from an evidence summary from the National Institute of Health and Clinical Excellence (NICE).

However, given the context we have revised the sentence (Page 6; line 135): There is insufficient evidence addressing effectiveness and safety for use of iodine to treat or prevent wound infection

Reviewer 2: Why have you omitted flat-knitted stockings for good compression?

Response: We collected data on any stocking use.

Reviewer 2: Pentoxifylline is not recommended for wound management in norther Europe and seems to be one agent that there is no consensus about. The reference is rather old (from 2010). Do you have further, more recent references, to back up your statement. In box 1, you clearly mention that this agent is believed to increase microcirculatory blood flow and should be considered in patients with VLU. Please elaborate your statement or simply omit this agent.

Response: In the UK, pentoxifylline is recommended for use as a VLU treatment in SIGN guidelines, which are nationally recognised. We are not making judgements on this recommendation but are rather presenting the recommendation of an expert guidelines panel and relating it to current use in a large region of the UK. The lack of use in relation to the recommendation is, we feel, a very interesting finding that requires highlighting and attention. Reference 37 is national and well respected Guideline, additionally Reference 39 is a Cochrane review. We have added Reference 39 and text to box 1 to highlight the quality of this evidence (Page 6; line 135): High quality evidence based on systematic review and meta-analysis has found improved VLU healing with the use of pentoxifylline. Since this systematic review was published we are only aware of one further relevant RCT which has 40 participants and seems of very low quality. This would not impact on the findings of the review. Because of the points above we hope the reviewer agrees there is value in leaving Box 1 unchanged.

Reviewer 2: A major concern is your statements about antimicrobial dressings and not relating these dressings to the overuse of oral antibiotic treatment, which is far more harmful to both the patients and society as a whole. I think you should relate these topics to one another and discuss this matter in detail.

Response: Overuse of oral antibiotics is a concern across health systems. However, our study was a multi-service cross-sectional survey that took a snapshot, over a two week period, of current practices in wound assessment, prevention and treatment. We did not investigate overuse of oral antibiotic treatment generally. As such, any comment would be speculation on our part and not grounded in the data presented.

Reviewer 2: "To reduce duplication, discussions took place with services providing shared care to minimise the chance of data being collected twice for the same person". How exactly did you do that?

Response: We have added text to explain how we did this (Page 7; line 161): To reduce duplication, forms were completed by the person providing the most hands-on care. This was supported by further local processes, such as placing stickers in patients' clinical notes when a form had been completed. Potential duplicates were not removed from the dataset, as data were anonymised we could not be sure whether they were true duplications.

Reviewer 2: The fact that you did find more men than women is intriguing as women usually are overrepresented in this kind of patient population. It would be a good idea to elaborate this somewhat surprising finding in Discussion.

Response: We thank you for your close scrutiny of our manuscript and for highlighting that Table 1 showed that there were more men than women. We agree that this would have been an interesting finding. Our first reaction was to check the figures and in doing so we found an error in the labelling of gender in Table 1. We are not sure how we missed this and apologise for the oversight. We have

corrected Table 1 (Page 10; line 199): Male n (%) 1439 (49) Female n (%) 1528 (51). We have also checked all other figures throughout the manuscript. This was the only error, all other figures are correct.

Reviewer 2: There seems to be room for better presentation of the results, like presenting your finding of the gender perspective in your material.

Response: We have explained the reason for the error above.

Reviewer 2: I think you have to rewrite the paragraph on silver as your assumptions for calculations are completely wrong. In clinical practice it is recommended to use silver dressings for a short period of time (10-14 days) for local ulcer infections. Silver dressings should only be prescribed by a physician. Dressing changes are usually made 2 a week. This makes your calculations way to high and not at all reflecting clinical practice. My suggestions is that you rewrite the whole paragraph on silver as it is quite misleading.

Response: The costs have been amended the text to reflect the reviewer's views around dressing use (Page 12; line 257): Silver dressings cost several times more than their non-silver counterparts. In a conservative scenario where silver dressings are used for a patient over a 2-week period with two dressing changes per week, the dressing cost (using an average cost for a small dressing) would be approximately £32. The cost for a standard foam dressing (of the same size) in the same scenario is approximately five times less (£6.80). Use of silver dressings in this situation (assuming all other aspects of care and outcome are equal) corresponds to an additional £2,520 per 100 patients treated with silver dressings.

However, we have noted that this is likely a very conservative estimate for the UK where use is unlikely to be as contained in a two-week period for all patients. In the UK, wound dressings (including silver and other antimicrobial preparations) are mainly prescribed by experienced nurse prescribers. Physicians, such as general practitioners do prescribe dressings but usually only when a nurse prescriber is not managing the patients care. We have not changed other aspects of the paragraph as all other statements are supported by current evidence

Reviewer 2: Concerning compression therapy I do agree that there is much to be done within this field. I do miss a discussion about dressing changes, that is the use of the nurse's time, which stands for 87% of costs for wound management. With a proper compression therapy dressing changes could be done once a week, which very much reflects cost savings aspects. What is your comment on this?

Response: Thank you for this comment: we agree with the reviewer in this point, that better use of compression seems vital to maximise benefits in the care of those with VLUs - and this is stated widely in the manuscript; notably in Box 1. Future work might be an evaluation of improved care where we would measure nursing time and see if there is a reduction in this without impact on patient benefit.

Reviewer 2: The largest concern is about pentoxifylline, which is scarcely used in northern Europe. You state that..."There is also an apparent underuse of pentoxifylline, a xanthine derivative used to treat muscle pain in people with peripheral arterial disease, but also believed to increase microcirculatory blood flow in patients with VLU". I do not think that this substance should be part of this article at all (only one reference from 2010). I advise you to omit this agent from the article.

Response: Please see earlier comment on this. We have explored use in relation to a Guideline recommendation. We are not judging further than this but exploring the current use and potential for savings using current data. Discussion of this issue seems important for the area so we would request that this remains. As noted before, the 2010 reference (Reference 37) is a national and well respected guidelines and the data are based on a Cochrane systematic review (Reference 38).

Reviewer 2: I do agree that in all cases, further work is required to identify the factors that underpin clinical decision making and behaviour in these areas. Opposed to you I do think that giving the patient a proper diagnosis and hence an appropriate treatment strategy would be a better way to

achieve improvement and thus value in healthcare. This includes a shift of paradigm in wound management not based on dressing materials.

Response: We thank the reviewer for this. We do not agree we have opposing views with the reviewer at all. We note firstly that the focus on dressings is only part of the paper. It is true to say that antimicrobial dressing use lacks supporting evidence. It seems useful to highlight the variation in practice and explore this issue. The emphasis on dressing material is concerned with reducing overuse of expensive dressings which lack evidence to release resources that can be used on more effective treatments (e.g. compression). Secondly we wish to flag the focus on measurement of ABPI and compression use in those with VLU which is part of the treatment strategy (as well as not being dressing related). Likewise use of pressure relieving surfaces for prevention in high risk patients speaks directly to assessment and treatment strategies. Indeed all of the issues that are flagged are inherently a wider part of issues around diagnosis and appropriate treatment. Returning to earlier points made – we see this paper as part of wider research that is required in the area to understand what current practice is, how this relates to local and national guidelines, the implications of current care and how improvements can be made.

VERSION 2 – REVIEW

REVIEWER	Rut F Öien Blekinge Center of Competence, Karlskrona, Sweden Faculty of Medicine, Lund University, Malmö, Sweden
REVIEW RETURNED	16-Jan-2018

GENERAL COMMENTS	Thank you for the amendments in your article presenting the results from a survey with data linked to guidelines. To promote the guidelines the authors have taken the use of antimicrobial dressings out of the context of clinical practice and present economic incentives like the costs for silver dressings compared with the costs for “conventional” dressings to state their argument. However, the use of antimicrobial dressings in clinical practice is one way to prevent irrational oral antibiotic treatment, which is a much larger threat to both the patients and society as a whole. It is clear that the use of antimicrobial dressings must be discussed in this context. I do urge the authors to take this into consideration and add this point of view in Discussion in order to balance the article and make it trustworthy.
--

VERSION 2 – AUTHOR RESPONSE

Comment Reviewer 2:

Thank you for the amendments in your article presenting the results from a survey with data linked to guidelines.

To promote the guidelines the authors have taken the use of antimicrobial dressings out of the context of clinical practice and present economic incentives like the costs for silver dressings compared with the costs for “conventional” dressings to state their argument.

However, the use of antimicrobial dressings in clinical practice is one way to prevent irrational oral antibiotic treatment, which is a much larger threat to both the patients and society as a whole. It is

clear that the use of antimicrobial dressings must be discussed in this context. I do urge the authors to take this into consideration and add this point of view in Discussion in order to balance the article and make it trustworthy.

Authors' Response

Thank you for this important point, we agree that systemic antibiotic overuse poses a global threat to public health. We are not aware, however, of convincing evidence that promotes the use of antimicrobial dressings as a way of reducing systemic antibiotic use or of reliable evidence that antimicrobial dressings can reduce bacterial infection. We have amended the findings and discussion sections to include systemic antibiotics to balance the argument and clarify our view and have added references to support this view.1-3

1. NICE. Chronic wounds: advanced wound dressings and antimicrobial dressings: Evidence summary [ESMPB2]: National Institute for Health and Care Excellence; 2016 [Available from: <https://www.nice.org.uk/guidance/esmpb2/resources/chronic-wounds-advanced-wound-dressings-and-antimicrobial-dressings-1502609570376901> (accessed November 2017).
2. O'Meara S, Al-Kurdi D, Ologun Y, et al. Antibiotics and antiseptics for venous leg ulcers. Cochrane Database Syst Rev 2014(1):CD003557. doi: 10.1002/14651858.CD003557.pub5
3. WHO. Antibiotic Resistance: Multi-country public awareness survey: World Health Organization, 2015.

Changes to manuscript text highlighted in blue (page 10 lines 203-205)

It is highly uncertain that antimicrobial dressings are clinically or cost effective and there is no high quality evidence that they improve wound healing, reduce infection rates or reduce the prescribing of systemic antibiotics.

Changes to manuscript text highlighted in blue (page 12 lines 250-253)

Whilst one could postulate that use of anti-microbial dressings might reduce the prescribing of systemic antibiotics (which would be highly desirable in the context of antimicrobial resistance) there is no evidence that this is the case. To examine the use of anti-microbial dressings in more depth we considered silver dressings for which the current evidence on the relative effectiveness for preventing or treating wound infection and promoting healing is uncertain.

Yours Sincerely

Trish Gray (corresponding/first author)
Research Fellow